# Apolipoprotein-AI mimetic peptides D-4F and L-5F decrease hepatic inflammation and increase insulin sensitivity in C57BL/6 mice

**Kristine C. McGrath**[1]*, **Xiaohong Li**[2], **Stephen M. Twigg**[3], **Alison K. Heather**[4,5]

**1** School of Life Sciences, University of Technology Sydney, Broadway, NSW, Australia, **2** Health Management Center, Shandong Provincial Hospital Affiliated to Shandong University, Jinan, Shandong, China, **3** Sydney Medical School (Central) and Charles Perkins Centre, Faculty of Medicine and Health, University of Sydney, NSW, Australia, **4** Department of Physiology, Otago School of Medical Sciences, University of Otago, Dunedin, New Zealand, **5** Heart Otago, University of Otago, Dunedin, New Zealand

* kristine.mcgrath@uts.edu.au

**Data Availability Statement:** All relevant data are within the paper.

**Funding:** This work was supported by University of Technology Sydney Chancellor's Postdoctoral

## Abstract

### Background

Apolipoprotein-AI (apo-AI) is the major apolipoprotein found in high density lipoprotein particles (HDLs). We previously demonstrated that apo-AI injected directly into high-fat diet fed mice improved insulin sensitivity associated with decreased hepatic inflammation. While our data provides compelling proof of concept, apoA-I mimetic peptides are more clinically feasible. The aim of this study was to test whether apo-AI mimetic peptide (D-4F and L-5F) treatment will emulate the effects of full-length apo-AI to improve insulin sensitivity.

### Methods

Male C57BL/6 mice were fed a high-fat diet for 16 weeks before receiving D4F mimetic peptide administered via drinking water or L5F mimetic peptide administered by intraperitoneal injection bi-weekly for a total of five weeks. Glucose tolerance and insulin tolerance tests were conducted to assess the effects of the peptides on insulin resistance. Effects of the peptides on inflammation, gluconeogenic enzymes and lipid synthesis were assessed by real-time PCR of key markers involved in the respective pathways.

### Results

Treatment with apo-AI mimetic peptides D-4F and L-5F showed: (i) improved blood glucose clearance (D-4F 1.40-fold AUC decrease compared to HFD, *P*<0.05; L-4F 1.17-fold AUC decrease compared to HFD, *ns*) in the glucose tolerance test; (ii) improved insulin tolerance (D-4F 1.63-fold AUC decrease compared to HFD, *P*<0.05; L-5F 1.39-fold AUC compared to HFD, *P*<0.05) in the insulin tolerance test. The metabolic test results were associated with (i) decreased hepatic inflammation of SAA1, IL-1β IFN-γ and TNFα (2.61–5.97-fold decrease compared to HFD, *P*<0.05) for both mimetics; (ii) suppression of hepatic mRNA expression of gluconeogenesis-associated genes (PEPCK and G6Pase; 1.66–3.01-fold decrease compared to HFD, *P*<0.001) for both mimetics; (iii) lipogenic-associated genes,

Research Fellowship Scheme (author KM) and the Diabetes of Australia Research Trust (authors KM, ST, AH). The funders had no role in study design, data collection and analysis, decision to publish, or preparation of the manuscript.

**Competing interests:** The authors have declared that no competing interests exist.

(SREBP1c and ChREBP; 2.15–3.31-fold decrease compared to HFD, $P<0.001$) for both mimetics and; (iv) reduced hepatic macrophage infiltration (F4/80 and CD68; 1.77–2.15-fold compared to HFD, $P<0.001$) for both mimetics.

## Conclusion

Apo-AI mimetic peptides treatment led to improved glucose homeostasis. This effect is associated with reduced expression of inflammatory markers in the liver and reduced infiltration of macrophages, suggesting an overall suppression of hepatic inflammation. We also showed altered expression of genes associated with gluconeogenesis and lipid synthesis, suggesting that glucose and lipid synthesis is suppressed. These findings suggest that apoA-I mimetic peptides could be a new therapeutic option to reduce hepatic inflammation that contributes to the development of overnutrition-induced insulin resistance.

## Introduction

The discovery of the anti-inflammatory and antioxidant properties of high density lipoproteins (HDL) has led to the question of whether HDL can be used therapeutically to treat inflammation in disease. Insulin resistance is critically dependent on liver inflammation with studies showing that constitutively active hepatic nuclear factor-kappaB (NF-κB), the central mediator driving the inflammatory response, leads to an insulin resistant state in a mouse model [1]. We previously demonstrated that administration of apolipoprotein A-I (apoA-I), either independently (lipid free) or as part of a reconstituted HDL with (rHDL) containing PLPC protects against liver inflammation and improves insulin resistance in high-fat fed C57Bl/6 mice [2]. Whilst the results are promising, the generation of apoA-I particles are large and time consuming.

Apolipoprotein A-I (apoA-I) mimetic peptides have been developed and designed to function similarly to full length apoA-I, with greater potency and a better pharmacokinetic profile than full length apoA-I [3,4]. These mimetic peptides offer many advantages over rHDL including the relative low cost, simplicity of production and the ability to modify their structure to allow oral administration. ApoA-I mimetic peptides are characterized by their phenylalanine residue attachments, with an increase in phenylalanine residue correlating with increasing hydrophobicity and their ability to associate with phospholipids [4]. Based on the number of hydrophobic phenylalanine residues in the sequence, the most well studied are the 4F and 5F apoA-I mimetic peptides. These have the same class A amphiphatic helical structure as apoA-I [5]. D-4F and L-5F differ from each other with respect to the number of phenylalanines present on the hydrophobic face of the amphiphatic helix and that D-4F is synthesized from D-amino acids while L-5F is synthesized from L-amino acids. For oral administration, D-amino acids have been found to resist enzymatic degradation compared to L-amino acids. However, after absorption, D-amino acids would remain undegraded that can lead to toxicity and other side effects [6]. L-amino acids are less toxic in the circulation however they are not resistant to enzymatic degradation via oral administration. Both have been used previously in biological studies with reported potent anti-inflammatory and antioxidant effects [7–12].

In the present study we tested directly whether treatment with the apoAI mimetic peptides, D-4F and L-5F, can improve insulin sensitivity associated with decreased hepatic inflammation in high fat diet (HFD)-fed mice. We show that the mimetic treatments are remarkably

effective at decreasing hepatic inflammation and improving insulin sensitivity. The efficacy of the mimetic peptides is similar to what we have previously published for rHDLs [2].

## Materials and methods

### Mimetic peptides

The N-terminal acetylated and C-terminal amidated apoA-I mimetic peptides were purchased from ChinaPeptides (Shanghai, China). The "4F" denotes the peptide: Ac-D-W-F-K-A-F-Y-D-K-VA-E-K-F-K-E-A-F-NH$_2$ synthesized from D-amino acids (D-4F). The "5F" denotes the peptide: Ac-D-W-L-K-A-F-Y-D-K-V-F-E-K-F-K-E-F-F-NH$_2$, synthesized from L-amino acids. Lyophilized peptide was stored at -20˚C and solubilized in sterile saline immediately prior to use.

### Animal model of high fat diet-induced insulin resistance

To prove the concept, males were used in this study as they are not affected by periodical changes in sex hormones. Forty 6-week old male C57Bl/6 mice were obtained from the Australian BioResources (Moss Vale, NSW, Australia) and housed in a 12-hour light/12-hour dark cycle with food and water available *ad libitum*. Of the 40 mice, 10 were fed a normal chow diet (12% fat; 14 MJ/kg, Specialty Feeds, NSW, Australia), and 30 mice were fed a high fat diet (43% fat; 20 MJ/kg, SF03-020, Specialty Feeds, WA, Australia). The HFD has been repeatedly used to induce obesity in rodents by us [2,13]. At 16 weeks of age, when all HFD-fed mice had established insulin resistance, the mice were subdivided with 10 subsequently treated with D-4F (300 µg/mL [14] in drinking water), another 10 were treated with L-5F (10 mg/kg [14] i.p injection twice per week) and another 10 were administered endotoxin-free saline as the vehicle control until 21 weeks of age. Body weights were recorded weekly. All animals were administered their last peptide dose 24 hr before euthanasia by cardiac puncture after full anaesthesia has been achieved using isoflurane. At sacrifice, serum was collected and stored at –20˚C. Liver, visceral and epididymal fat pads were then carefully excised, weighed, snap frozen in liquid nitrogen and stored at –80˚C. All animal experiments were conducted in accordance to the guidelines described by the Australian National Health and Medical Research council code of conduct for animals with approval from the University of Technology Sydney Animal Care and Ethics Committee, Sydney, NSW, Australia (ACEC#ETH11-442A).

### Glucose and insulin tolerance test

Intraperitoneal (i.p.) glucose tolerance tests (GTT) and intraperitoneal insulin tolerance tests (ITT) were performed in overnight fasted mice, as previously described [2]. Briefly, animals were injected i.p. with 2 g/kg of D-Glucose (Sigma Aldrich, St Louis, MO, USA) for the GTT or with 0.5 IU/kg of insulin (Humulin, Lilly, France) for the ITT. Blood glucose levels were measured (Accu-Chek, Roche, Castle Hill, NSW, Australia) at indicated times. The area under the curve (AUC) of glucose levels was calculated for each mouse.

### Insulin and lipid analysis

Serum insulin levels were measured by ELISA (Crystal Chem, Downers Grove, IL, USA). Serum and tissue triglyceride levels were measured using triglyceride reagent (Roche Diagnostics, Castle Hill, NSW, Australia) after lipids were extracted from tissue using a mixture of chloroform and methanol (2:1, v:v) as previously described [2]. Serum NEFA levels were measured using a commercially available kit (WAKO, Osaka, Japan).

### RNA isolation and RT-qPCR

Total RNA was isolated from tissue using TRIsure (Bioline, Eveleigh, NSW, Australia) and bioanalysis was done using the Experion system (Bio-Rad, Hercules, CA). cDNA was reverse transcribed from total RNA (200 ng) using the SensiFAST cDNA synthesis kit (Bioline) and gene expression determined by qPCR (CFX96, Bio-Rad), using specific primers described previously [2]. Relative change in mRNA gene expression was determined as described previously, using ubiquitin C (UBC) as the reference gene (3). The average value of the control was assigned as the calibrator, against which all other samples were expressed as percentage difference.

### Statistical analysis

All values are reported as mean ± standard error of mean (SEM). Significant differences in treatments was determined by one-way ANOVA with Bonferroni's posthoc test using GraphPad Prism 6 (La Jolla, CA, USA). $P < 0.05$ was considered significant.

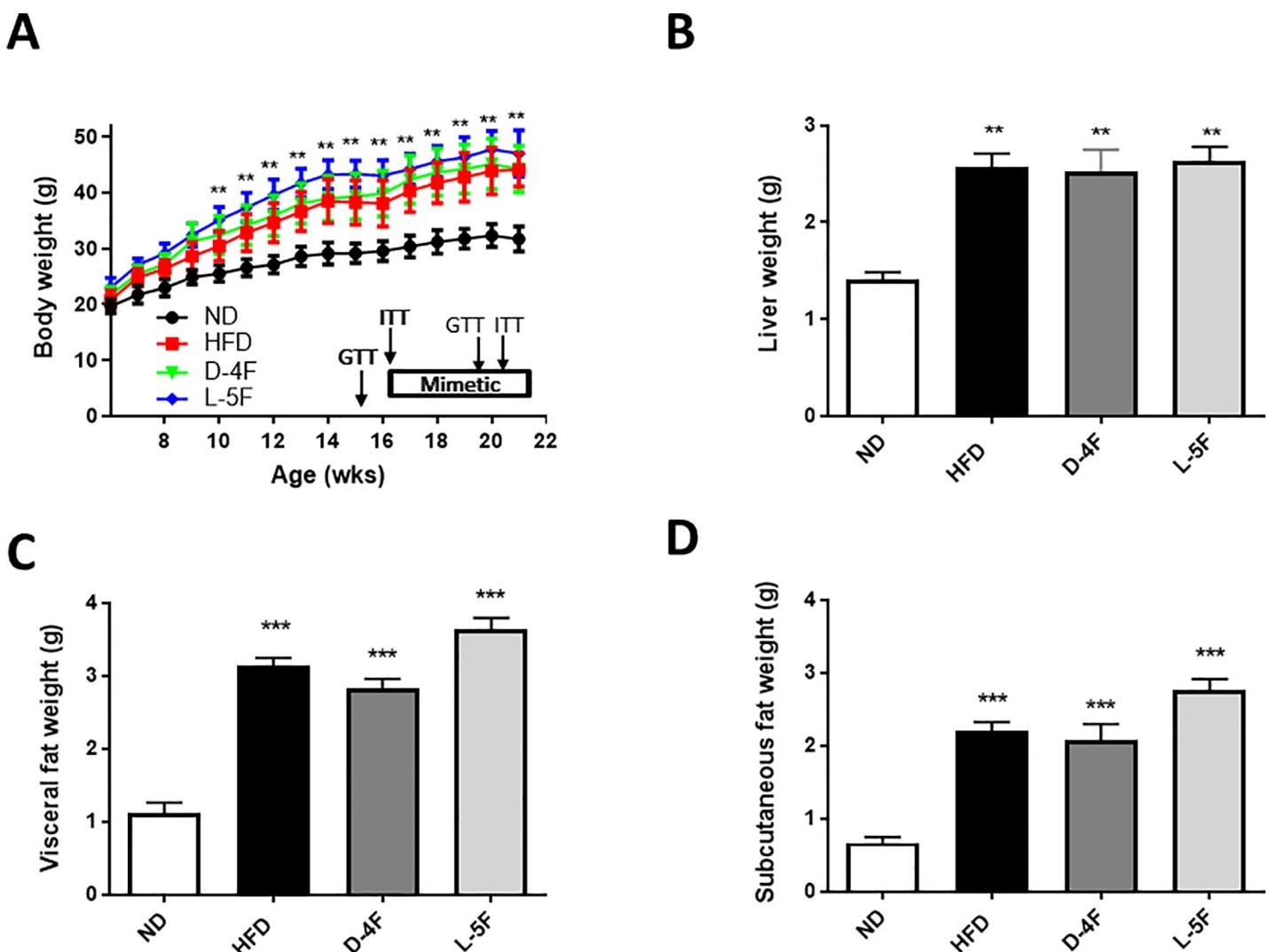

**Fig 1. D-4F and L-5F did not affect body, liver and adipose tissue weight gain.** Weight of body (A), liver (B), visceral fat (C) and subcutaneous fat (D). Results are mean ± SEM (n = 8–10). $^*P < 0.05$ vs. ND; $^*P < 0.001$ vs. ND; $^*P < 0.0001$ vs. ND.

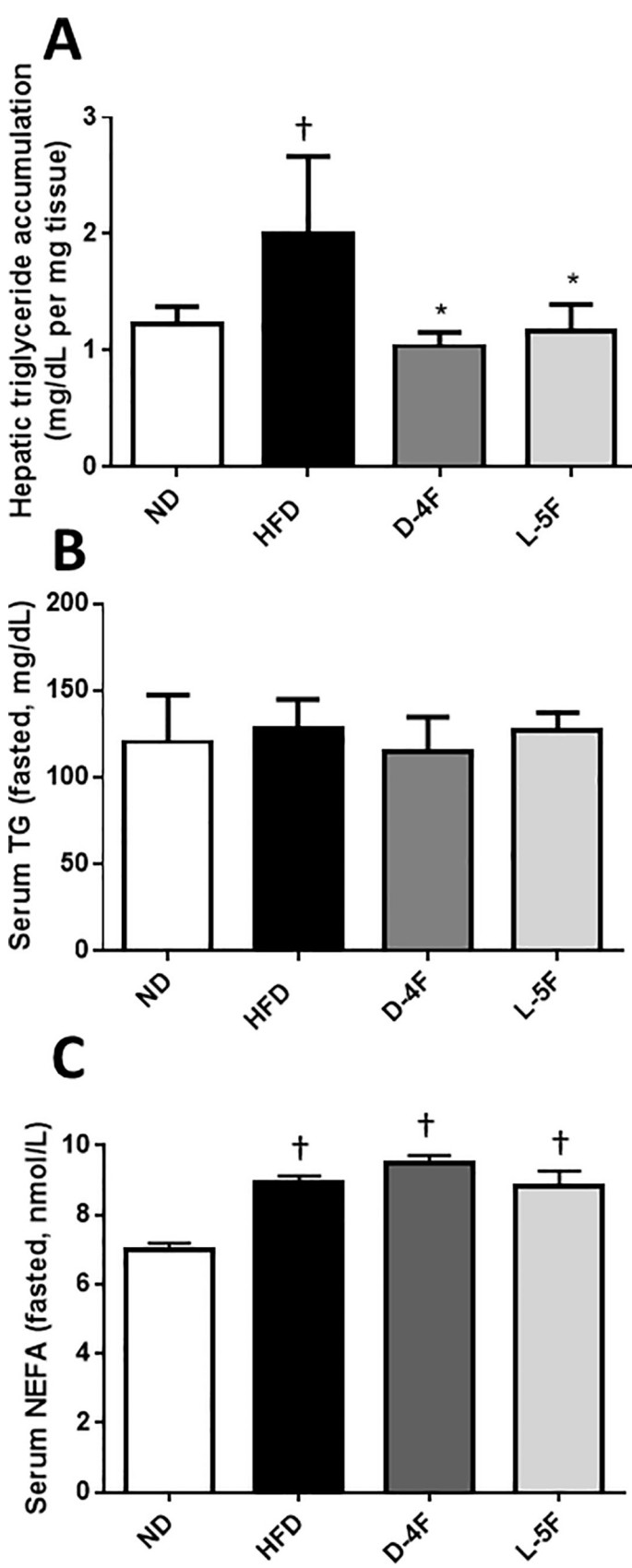

**Fig 2. D-4F and L-5F attenuate hepatic triglyceride accumulation.** Hepatic triglyceride (A), serum triglyceride (B) and serum NEFA (C). Results are mean ± SEM (n = 8–10). †$P$<0.05 vs. ND; *$P$<0.05 vs. HFD.

## Results

### D-4F or L-5F did not attenuate weight gain in high fat diet-fed C57BL/6 mice

In this study, C57Bl/6 mice were fed a standard chow-diet (ND) or a high fat diet (HFD) *ad libitum* that was intended to induce a large weight gain, dyslipidemia and insulin resistance [15]. After 10-weeks on the diet, HFD-fed mice were randomly assigned to apolipoprotein AI mimetic peptides, D-4F or L-5F, treatment. Fig 1A shows the time course of body weight gain during the 16 weeks of HFD with or without D-4F or L-5F treatment. As expected, after 16 weeks, body weight was significantly increased in all mice on HFD compared with ND ($P$<0.001). The increase in body weight was associated with increased liver weight ($P$<0.001; Fig 1B), visceral fat weight ($P$<0.0001; Fig 1C) and subcutaneous fat weight ($P$<0.0001; Fig 1D). Treatment with D-4F or L-5F peptides did not reduce the HFD-induced body, liver or adipose tissue weight gain.

### D-4F and L-5F attenuate hepatic triglyceride accumulation

The increase in body weight, liver weight and fat mass in response to the HFD was associated with increased hepatic triglycerides compared to normal controls ($P$<0.05; Fig 2A), however both D-4F and L-5F treatment brought triglyceride levels back to baseline. The increase in hepatic triglyceride levels did not result in systemically high triglycerides (Fig 2B), however it was associated with an increase in serum NEFA by 22±13% ($P$<0.05; Fig 2C), an effect that was not blocked by either D-4F or L-5F.

Excess dietary carbohydrates are converted to triglycerides in the liver through the pathway of *de novo* lipogenesis for long-term energy storage. Enzymes involved in *de novo* lipogenesis are induced by key transcriptional regulators including sterol regulatory element binding protein 1c (*SREBP-1c*) and carbohydrate responsive element binding protein (*ChREBP*). In agreement with the increased hepatic triglyceride levels in HFD-fed mice, *SREBP-1c* and *ChREBP* mRNA levels increased in these mice ($P$<0.001). Both D-4F and L-5F treatment decreased

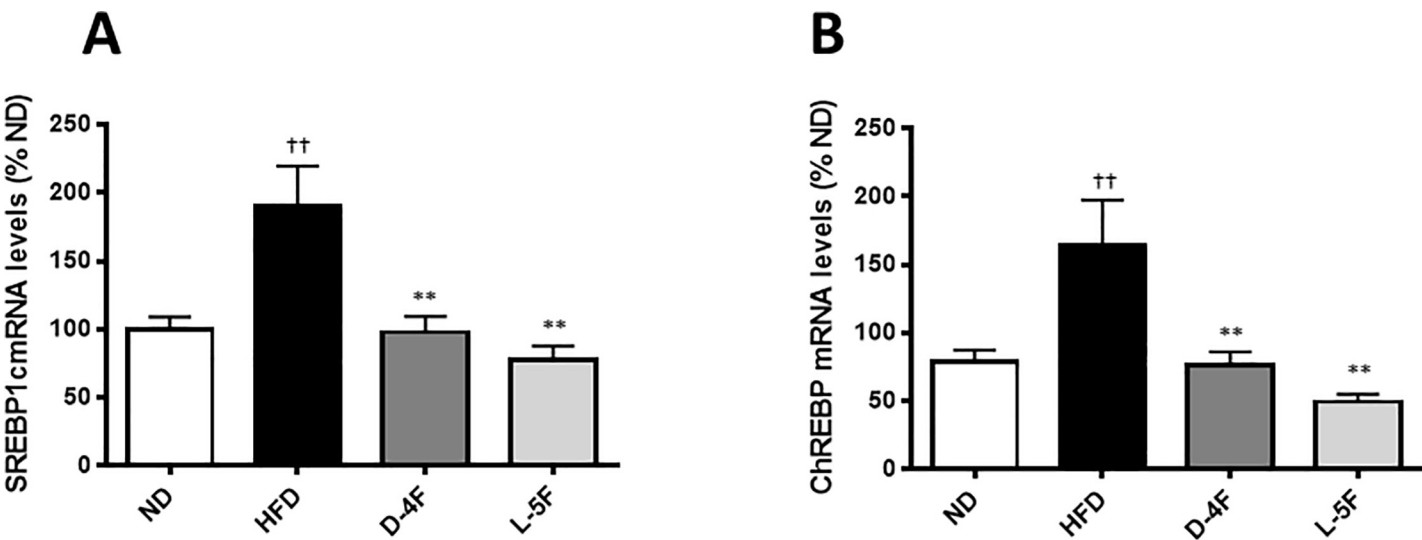

**Fig 3. D-4F and L-5F decrease hepatic mRNA levels of genes involved in lipogenesis.** Quantitative real-time PCR of key genes involved in lipogenesis (A) SREBP-1c and (B) ChREBP. mRNA levels were normalized to Ubiquitin C (UBC). Results are mean ± SEM (n = 8–10). †$P$<0.001 vs. ND; **$P$<0.001 vs. HFD.

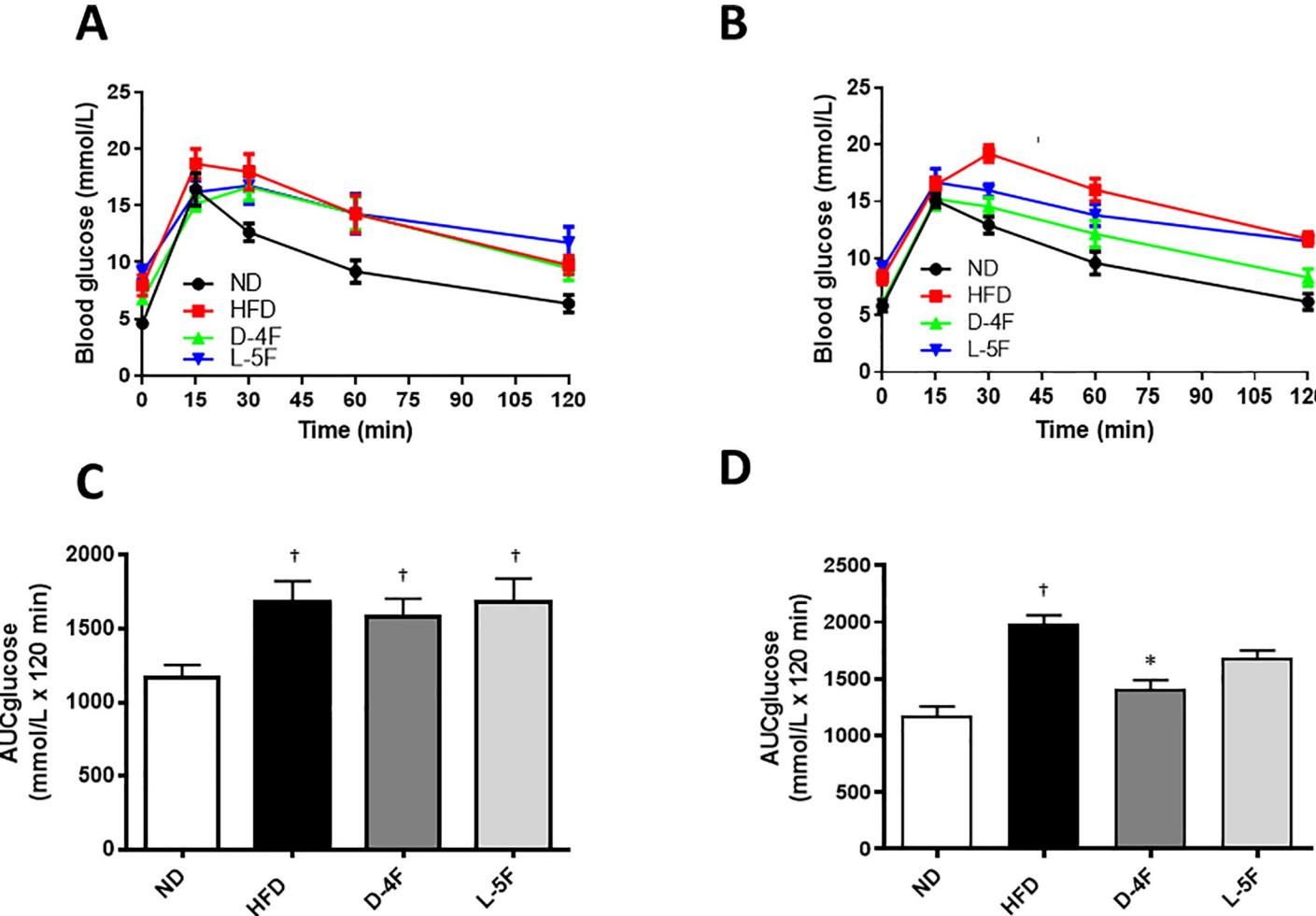

**Fig 4. D-4F and L-5F improves blood glucose.** Beginning at 6 weeks of age, C57BL/6 mice were fed a standard chow diet (ND) or high-fat diet for a total of 16 weeks. After 10 weeks, the high fat diet group was subdivided: HFD+endotoxin-free saline (HFD); HFD+D-4F (D-4F) or HFD+L-5F (L-5F) for the last 6 weeks of diet. Plasma glucose concentrations during intraperitoneal glucose tolerance test (GTT; glucose 2g/kg) (A) before treatment commenced and (B) after 4 weeks of treatment with the mimetic peptides. (C and D) Area under the curve for glucose (AUC glucose) was calculated using the trapezoid rule. Results are mean ± SEM (n = 8–10). †$P<0.05$ vs. ND; *$P<0.05$ vs. HFD.

HFD-induced *SREBP-1c* ($P<0.001$) and *ChREBP* ($P<0.001$) mRNA levels. This observation suggests that mimetic treatment suppressed HFD-induced lipogenesis (Fig 3A and 3B).

### D-4F and L-5F effects on glucose- and insulin tolerance

To determine the glucose tolerance of HFD-fed mice before treatment started with D-4F or L-5F, a GTT was performed. During the GTT, there was a significant increase in plasma glucose levels in HFD- vs. ND- fed mice (Fig 4A and 4C). Treatment with D-4F significantly improved glucose clearance compared to the HFD group as shown by a decrease in AUC compared to HFD ($P<0.05$, Fig 4B and 4D). No significance was found for the AUC in the group treated with L-5F (Fig 4D).

To assess the amount of insulin secreted in response to glucose, serum insulin levels were measured at various time points during glucose tolerance testing. The results show hyperinsulinemia in the HFD and L-5F group compared to ND. Treatment with D-4F, but not L-5F, normalized insulin levels compared to the HFD group (Fig 5A and 5B).

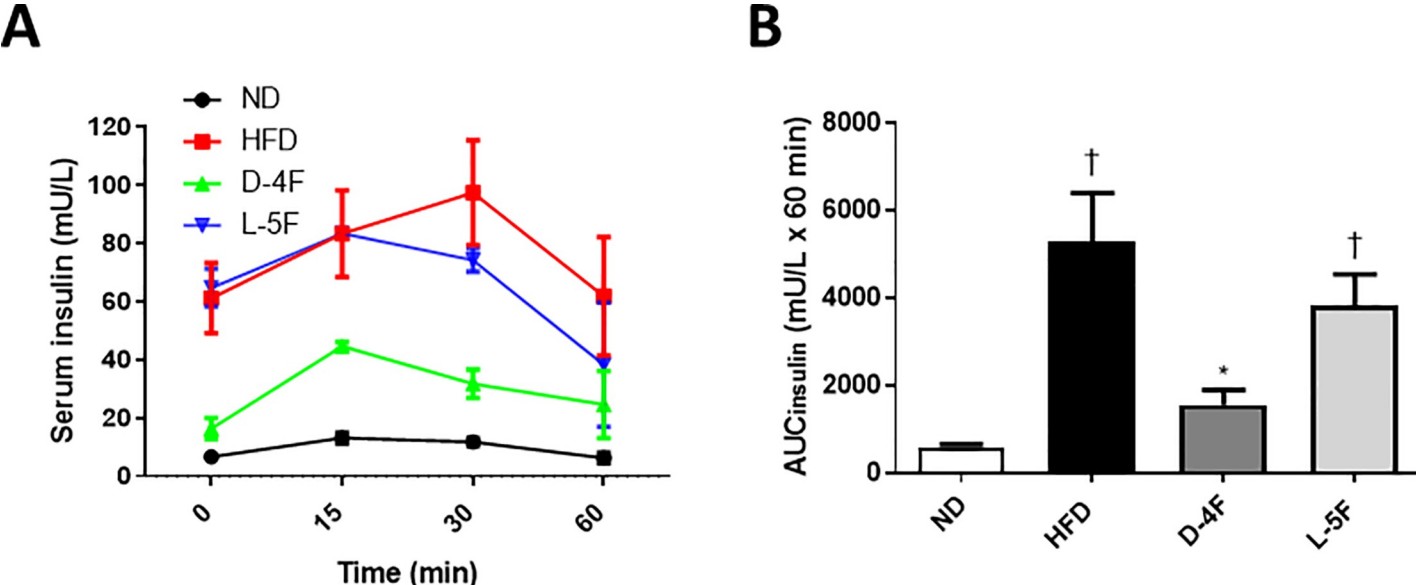

**Fig 5. D-4F normalizes insulin levels.** Beginning at 6 weeks of age, C57BL/6 mice were fed a standard chow diet (ND) or high-fat diet for a total of 16 weeks. After 10 weeks, the high fat diet group was subdivided: HFD+endotoxin-free saline (HFD); HFD+D-4F (D-4F) or HFD+L-5F (L-5F) for the last 6 weeks of diet. (A) Serum insulin levels were measured at various timepoints during the intraperitoneal glucose tolerance test (GTT; glucose 2g/kg). (B) Area under the curve for glucose (AUC glucose) was calculated using the trapezoid rule. Results are mean ± SEM (n = 4–6). †$P<0.05$ vs. ND; *$P<0.05$ vs. HFD.

Prior to treatment with D-4F or L-5F, insulin resistance of HFD-fed mice was assessed by an ITT. Insulin resistance was increased in the HFD group as shown by a higher AUC in all groups compared to ND (Fig 6A and 6C). Treatment with D-4F and L-5F improved insulin sensitivity ($P<0.05$, Fig 6B and 6D).

### D-4F and L-5F suppress hepatic mRNA levels of gluconeogenesis- and lipogenic associated genes

So far, our results, have indicated that D-4F and L-5F restores, at least in part, the disrupted glucose and insulin homeostasis induced by a HFD. It has previously been established that a HFD disrupts liver gluconeogenesis [16]. We, therefore, next tested if HFD affected the expression of the gluconeogenesis rate-limiting enzymes, phosphoenolpyruvate carboxykinase (PEPCK) and glucose-6-phosphatase (G6Pase), and whether D-4F and L-5F treatment alleviated any HFD effect. Our findings show that indeed HFD increased PEPCK ($P<0.05$ vs. ND) and G6Pase ($P<0.001$ vs ND) expression and that both D-4F and L-5F restored PEPCK and G6Pase levels back to baseline. These findings for PEPCK and G6Pase are in keeping with their effect on improved glucose tolerance (Fig 7A and 7B).

### D-4F and L-5F suppress hepatic inflammation

We have recently published that the improvement in insulin sensitivity in HFD-fed mice treated with rHDLs was associated with decreased hepatic inflammation [2]. We next tested whether D-4F and L-5F treatment could mimic this beneficial effect of rHDLs and also decrease hepatic inflammation as evidenced by decreased expression of key inflammatory markers, namely the cytokines, SAA1, IL-1β, IFN-γ and TNF-α. Fig 8 show that HFD significantly increased the expression of all four cytokines (IL-1β and IFNϒ, $P<0.05$; SAA1 and TNFα, $P<0.001$ vs ND); an effect that was abrogated by both D-4F and L-5F ($P<0.001$—$P<0.0001$ vs. HFD).

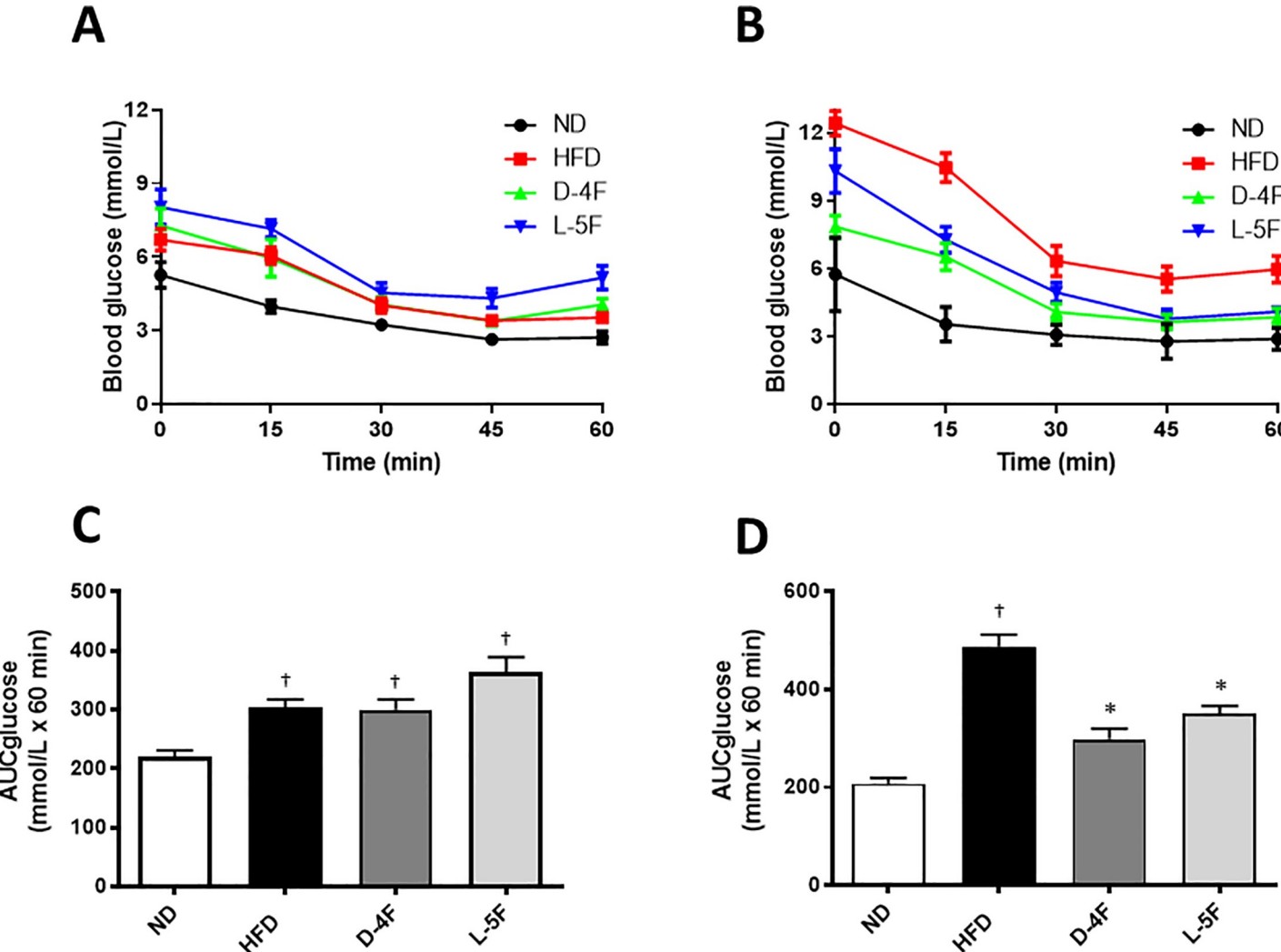

**Fig 6. D-4F and L-5F improves insulin sensitivity.** Beginning at 6 weeks of age, C57BL/6 mice were fed a standard chow diet (ND) or high-fat diet for a total of 16 weeks. After 10 weeks, the high fat diet group was subdivided: HFD+endotoxin-free saline (HFD); HFD+D-4F (D-4F) or HFD+L-5F (L-5F) for the last 6 weeks of diet. Plasma glucose concentrations during intraperitoneal insulin tolerance test (ITT; insulin 0.5 IU/kg) (A) before treatment commenced and (B) after 5 weeks of treatment with the mimetic peptides. (C and D) Area under the curve for glucose (AUC glucose) was calculated using the trapezoid rule. Results are mean ± SEM (n = 8–10). †$P < 0.05$ vs. ND; *$P < 0.05$ vs. HFD.

### D-4F and L-5F suppress hepatic macrophage infiltration

The involvement of hepatic macrophages or Kupffer cells in the pathogenesis of insulin resistance has been previously reported [1,16]. We next examined macrophage-specific gene expression levels in the liver as markers of macrophage infiltration. Our results show mRNA levels of *F4/80* (also known as *Emr1*) and *CD68* were upregulated in HFD-mice ($P < 0.001$; Fig 9). The expression of both genes were decreased by both D-4F ($P < 0.001$) and L-4F ($P < 0.001$).

## Discussion

Previous studies have revealed that chronic inflammation in hepatocytes is sufficient to drive the onset of insulin resistance, and that treatment of HFD-fed mice with reconstituted HDLs rescued insulin resistance associated with suppressed hepatic inflammation [1,2]. Despite the

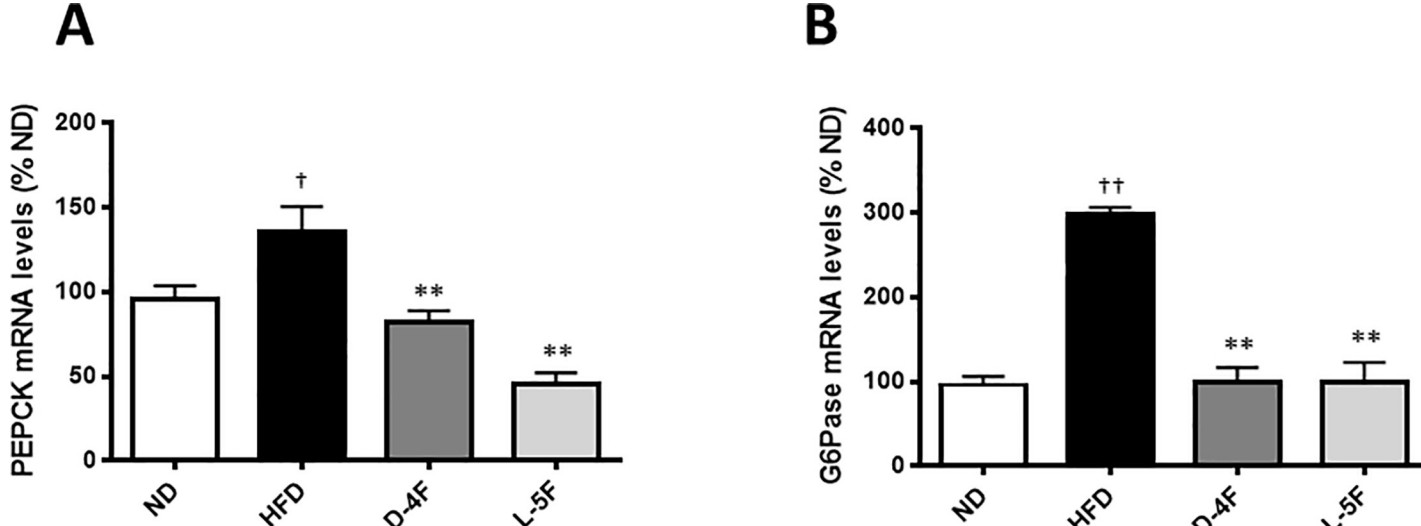

**Fig 7. D-4F and L-5F decrease hepatic mRNA level of genes involved in gluconeogenesis.** RT-qPCR of gluconeogenic enzymes (A) PEPCK and (B) G6Pase. mRNA levels were normalized to Ubiquitin C (UBC). Results are mean ± SEM (n = 8–10). †$P<0.05$ vs. ND; ††$P<0.001$ vs. ND; **$P<0.001$ vs. HFD.

successful treatment with rHDLs, its use as a feasible (cost, ease of production and administration) long-term clinical therapeutic option is not great. In the current study, we investigated whether the apoA-I mimetic peptides, D-4F and L-5F, were effective against HFD-induced insulin resistance. We uncovered three important findings. The first is that treatment of mice twice a week for four-weeks with D-4F was effective against the development of HFD glucose intolerance, the latter which is recognized as a first step towards type 2 diabetes. Treatment of mice with the peptides was also sufficient to markedly improve insulin sensitivity and importantly, the degree of improvement was in keeping with what was reported by us for mice treated in the same manner with rHDLs [2].

A second important finding is that the five-week treatment with either D-4F or L-5F suppressed hepatic inflammation. Cai *et al* report that chronic NF-κB activation in the liver of C57Bl/6 mice drives the onset of insulin resistance, even in the absence of a HFD [1]. Conversely, mice that express the inhibitor of NF-κB, IκB, do not develop insulin resistance, even when fed a HFD [1]. The fact that both D-4F and L-5F reduce hepatic inflammation, as demonstrated by decreased inflammatory cytokine expression and macrophage infiltration, suggest that a reduction in local inflammation within the liver may serve as an underlying mechanism by which HDL-based therapies can improve insulin resistance.

The third important finding is that D-4F and L-5F decrease hepatic lipid accumulation. These findings suggest that D-4F and L-5F interfere with hepatic *de novo* lipid synthesis by directly decreasing SREBP1c levels, a transcription factor that can promote lipid production in the liver. This scenario is plausible because our findings show that both D-4F and L-5F decreased hepatic triglyceride levels, despite no change occurring in serum triglyceride concentrations. An additional observation that lends support to the notion that D-4F and L-5F might play a direct role in decreasing hepatic lipid accumulation is that D-4F and L-5F both decreased hepatic SREBP-1c and ChREBP expression, enzymes involved in *de novo* fatty acid biosynthesis.

In the present study, we used C57Bl/6 mice to investigate D-4F and L-5F effects on improving HFD-induced insulin resistance. Our findings in this study are in keeping with our previous reports for rHDLs and that of Peterson *et al* who demonstrated L-4F treatment improved insulin sensitivity and improved glucose tolerance in the *ob/ob* mouse model of diabetes [17].

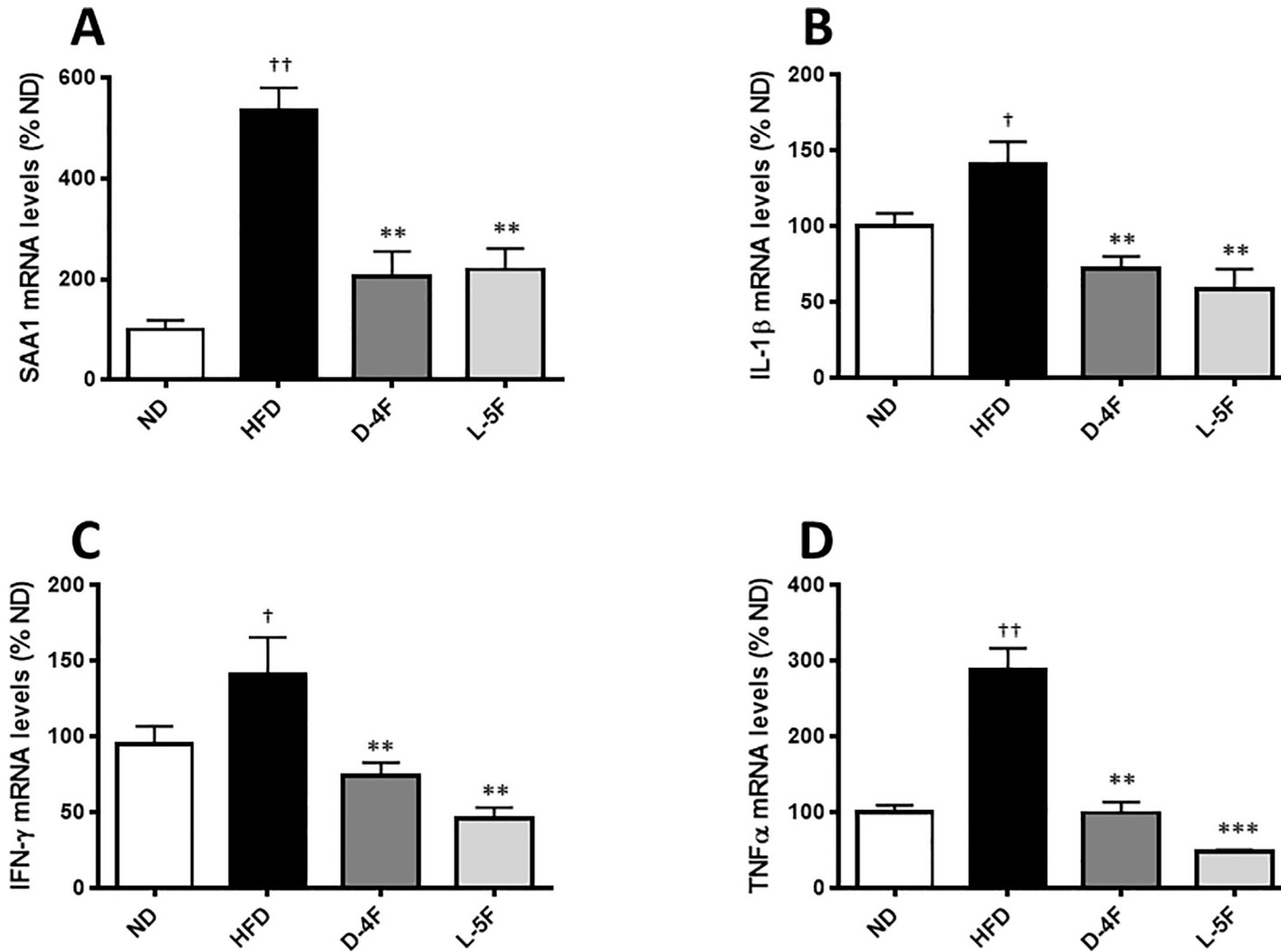

**Fig 8. D-4F and L-5F decrease hepatic mRNA level of genes involved in inflammation.** RTqPCR of genes involved in inflammation in the liver. mRNA levels were normalized to Ubiquitin C (UBC). Results are mean ± SEM (n = 8–10). †$P<0.05$ vs. ND; ††$P<0.001$ vs. ND; **$P<0.001$ vs. HFD; ***$P<0.0001$ vs. HFD.

Peterson *et al* showed that L-4F treatment increased serum adiponectin levels, which likely resulted in the reduction of fat content in the liver, that was a primary observation of their study [17]. In our study, we did not measure adiponectin levels, however no changes in fat content were noted for either D-4F or L-5F treatments.

Despite the positive effect that L-5F had on insulin resistance, it was not as effective at improving glucose tolerance when compared to D-4F, even though a trend towards improvement was observed. Insulin levels during glucose tolerance testing showed the cohort of mice treated with L-5F had hyperinsulinemia similar to that of HFD-fed mice despite L-5F showing improved insulin sensitivity in the insulin tolerance test. This suggests that endogenous insulin secreted during the GTT in the L-5F cohort is not as effective compared to the ND or the cohort treated with D-4F suggesting insulin receptor levels, or other metabolic parameters, are not restored by L-5F. The differential responses of D-4F and L-5F need further investigation to fully understand the differences in their mechanism of action. The strong effects of D-4F are interesting given previous studies that show that the bioavailability of this mimetic in plasma is poor (<1%, [18]). It has been proposed that D-4F exerts its effects via the binding of D4F to

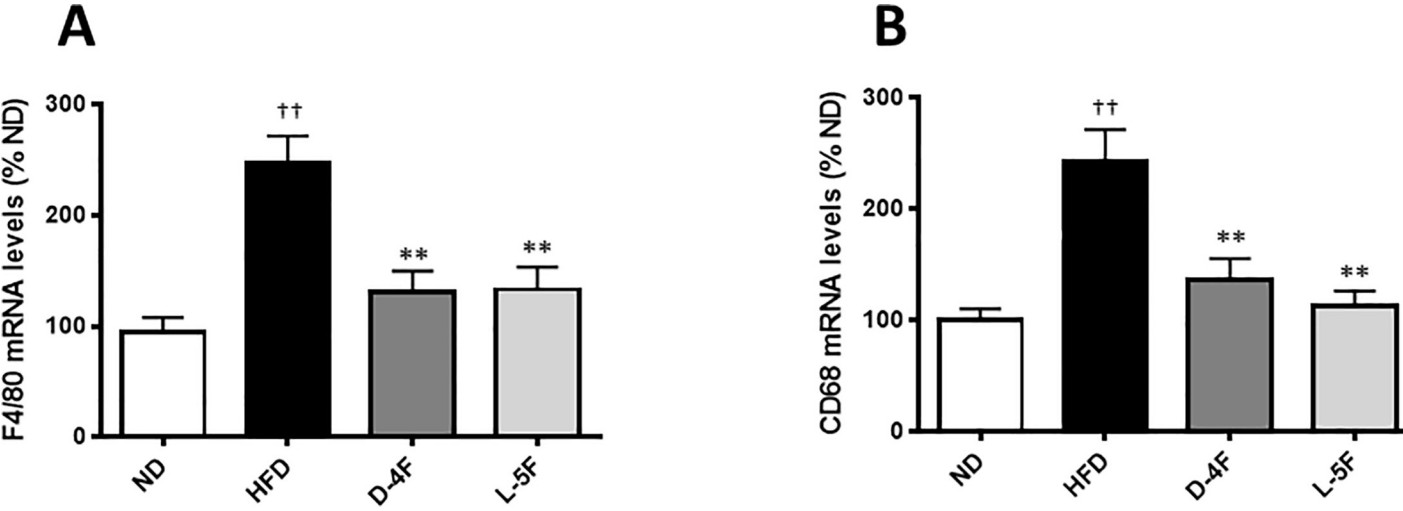

**Fig 9. D-4F and L-5F decrease hepatic mRNA levels of monocyte-macrophage markers.** RTqPCR of monocyte-macrophage markers in the liver. mRNA levels were normalized to Ubiquitin C (UBC). Results are mean ± SEM (n = 8–10). ††$P<0.001$ vs. ND; **$P<0.001$ vs. HFD.

oxidized lipids in the intestine. The D4F interaction with oxidized lipids leads to their inactivation ultimately associated with decreased systemic inflammation [19]. L5F is injected rather than ingested orally, and is therefore unable to induce anti-inflammatory effects in the intestine, with subsequent less effectiveness. Such intestine-led anti-inflammatory effects could help explain the divergent effects of D-4F and L-5F in our study.

While our findings for D-4F are in keeping with our previous reports for rHDLs [15] and Peterson et al [17], they contrast with those reported by Averill *et al* for L-4F in the *Ldlr*[-/-] mouse model [20]. In these mice, L-4F failed to improve insulin resistance or adipocyte inflammation. There are a number of differences that could explain the conflicting results including the form of 4F that was used. In our study, the D form that is stable for oral delivery was used while the L form was used in the *Ldlr*[-/-] mice and this requires injection. This suggests that there may be differences in bioavailability of D-4F vs L-4F/L-5F, which are working via a different mechanism of action. Another difference may be that C57Bl/6 mouse strains are more sensitive, and *Ldlr*[-/-] more resistant, to apoA-I mimetic peptides. There is evidence that dosages used in *Ldlr*[-/-] mice that have successfully suppressed inflammation were 9–25 times higher [21–23] than the dose used in the Averill *et al* study [20]. Therefore, future studies should consider using multiple models of insulin resistance (genetic and diet) to better characterize the protective capacity of apoA-I mimetic peptides against insulin resistance.

Reduced levels of circulating HDLs and apoA-I are very common in the diabetic population [24]. Whilst the Investigation of Lipid Level Management to Understand its Impact in Atherosclerotic Events (ILLUMINATE) trial failed to show improvements in cardiovascular disease, increased HDL levels as a result of treatment with torcetrapib reduced plasma glucose, insulin and IR by homeostatic model assessment in patients, suggesting increased HDL levels may improve IR [24]. The mechanisms accounting for the actions of increased HDLs on IR are multiple and include actions in the pancreas, skeletal muscle and liver, as shown by us and others [2,25,26]. In this study, the D-4F-induced improvement in glucose clearance associated with suppression of liver inflammation. Whilst reduced liver inflammation may well underlie the systemic effect of improved glucose clearance, it is very likely that improved glucose clearance is a combined effect of D-4F on increased insulin secretion from pancreatic β cells [27], increased uptake of glucose by skeletal muscle cells [28], decreased inflammatory responses in

coronary endothelium [29] and adipose tissue [30], as well as the decreased liver inflammation reported here.

In summary, our findings for the apoA-I mimetic peptides adds to the growing evidence supporting a strong relationship between apoA-I and HDL treatments and reduced inflammation, insulin resistance and type 2 diabetes in mouse models. Further research is now warranted to develop the best approach for targeting the HDL-based therapies to prevent or treat insulin resistance.

## Author Contributions

**Conceptualization:** Kristine C. McGrath, Alison K. Heather.

**Data curation:** Kristine C. McGrath, Xiaohong Li.

**Formal analysis:** Kristine C. McGrath.

**Funding acquisition:** Kristine C. McGrath.

**Investigation:** Kristine C. McGrath.

**Methodology:** Kristine C. McGrath.

**Project administration:** Kristine C. McGrath.

**Resources:** Kristine C. McGrath.

**Supervision:** Alison K. Heather.

**Validation:** Kristine C. McGrath.

**Writing – original draft:** Kristine C. McGrath.

**Writing – review & editing:** Kristine C. McGrath, Stephen M. Twigg, Alison K. Heather.

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
