## [Decision Letter · Decision Letter 0]

1 Oct 2019

PONE-D-19-23129

Apoplipoprotein-AI mimetic peptides D-4F and L-5F decrease hepatic inflammation and increase insulin sensitivity in C57BL/6 mice

PLOS ONE

Dear Dr McGrath,

Thank you for submitting your manuscript to PLOS ONE. After careful consideration, we feel that it has merit but does not fully meet PLOS ONE’s publication criteria as it currently stands. Therefore, we invite you to submit a revised version of the manuscript that addresses the points raised during the review process.

We would appreciate receiving your revised manuscript by Nov 15 2019 11:59PM. To enhance the reproducibility of your results, we recommend that if applicable you deposit your laboratory protocols in protocols.io, where a protocol can be assigned its own identifier (DOI) such that it can be cited independently in the future. For instructions see: http://journals.plos.org/plosone/s/submission-guidelines#loc-laboratory-protocols

We look forward to receiving your revised manuscript.

Kind regards,

Michael Bader

Academic Editor

PLOS ONE

**Journal Requirements:**

2. At this time, we request that you  please report additional details in your Methods section regarding animal care, as per our editorial guidelines:

(1) Please describe any steps taken to minimize animal suffering and distress, such as by administering analgesics

(2) Please include the method of euthanasia

Thank you for your attention to these requests.

**Comments to the Author**

1. Is the manuscript technically sound, and do the data support the conclusions?

Reviewer #1: Yes

Reviewer #2: Yes

2. Has the statistical analysis been performed appropriately and rigorously? 

Reviewer #1: Yes

Reviewer #2: Yes

3. Have the authors made all data underlying the findings in their manuscript fully available?

Reviewer #1: No

Reviewer #2: Yes

4. Is the manuscript presented in an intelligible fashion and written in standard English?

Reviewer #1: Yes

Reviewer #2: Yes

5. Review Comments to the Author

Reviewer #1: Manuscript Number: PONE-D-19-23129

Apoplipoprotein-AI mimetic peptides D-4F and L-5F decrease hepatic inflammation and increase insulin sensitivity in C57BL/6 mice by Kristine Ching-Yee McGrath et al.

The present manuscript has been described that D-4F and L-5F, the Apo-AI mimetic peptides treatment led to improved glucose homeostasis, and this effect is associated with reduced expression of inflammatory markers in the liver and reduced infiltration of macrophages, suggesting an overall suppression of hepatic inflammation.

These observations are interesting, and some specific point should be considered.

Specific comments

Authors have described only D-4F and L-5F as Apo-AI mimetic peptides, however various Apo-AI mimetic peptides have been developed so far. Authors should show about other peptides such as 5A, 18A, and FAMP...

In addition to Apo-AI mimetic peptides, we also have reconstituted-HDL such as ApoA-1/POPC disc. Author should describe the difference between Apo-AI mimetic peptides and reconstituted-HDL.

In this study, the authors conducted i.p.GTT as an assessment of glucose tolerance, but this method neglected the effects of incretin. Is it not necessary to consider the effect of incretin on the results of this study?

CETP inhibitor, one of the HDL-targeting drug also improves glucose homeostasis. The authors should consider the difference in the mechanism between the effect of CETP inhibitor and the effect of Apo-AI mimetic peptides.

HOMA-IR and HOMA-beta should be shown.

Fig. 1A; 4A; 4B; 5A; 6A; 6B, It is very difficult to understand group differences in the line graphs.

Reviewer #2: Interesting paper examining effect of apoA-I mimetic peptides (D4F and L5F) on glucose metabolism in C57BL6 mice fed a high fat diet.

1. Abstract: Quantitative description of key findings and P-values should be added.

2. Methods: Method used to quantify visceral and SQ fat should be stated. Time from. Last peptide dose for drawing plasma from mice should be stated.

3. Lipoprotein analysis: Ideally, besides plasma triglycerides, total cholesterol should also be analyzed. A more detailed analysis of plasma lipoproteins would also be helpful.

4. Discussion: Short paragraph should be added on potential mechanism of action of peptides in regard to observed effects. Oral D4F has been proposed to act via modulating oxidized lipid content in the intestine. This works should be cited and used to possibly explain effect of D4F and lack of effect of L5F. Should discuss how the magnitude of the effect on glucose metabolism by D4F compares to past studies on treatment with rHDL.

6. PLOS authors have the option to publish the peer review history of their article (what does this mean?). If published, this will include your full peer review and any attached files.

Reviewer #1: No

Reviewer #2: No

---

## [Author Response · Author response to Decision Letter 0]

5 Dec 2019

Reviewer #1: Manuscript Number: PONE-D-19-23129

Apoplipoprotein-AI mimetic peptides D-4F and L-5F decrease hepatic inflammation and increase insulin sensitivity in C57BL/6 mice by Kristine Ching-Yee McGrath et al.

The present manuscript has been described that D-4F and L-5F, the Apo-AI mimetic peptides treatment led to improved glucose homeostasis, and this effect is associated with reduced expression of inflammatory markers in the liver and reduced infiltration of macrophages, suggesting an overall suppression of hepatic inflammation.

These observations are interesting, and some specific point should be considered.

1. Authors have described only D-4F and L-5F as Apo-AI mimetic peptides, however various Apo-AI mimetic peptides have been developed so far. Authors should show about other peptides such as 5A, 18A, and FAMP...

Response: With so many different mimetic peptide sequences exhibiting various degrees of activity, it is not possible to select lead sequences with confidence. Whilst it would be of great interest to investigate the effects of the other peptides, it would be challenging to include all of them in the one study. To address the reviewer’s comment, we have now clarified this in the introduction section. Please see page 4, line 75-79.

2. In addition to Apo-AI mimetic peptides, we also have reconstituted-HDL such as ApoA- 1/POPC disc. Author should describe the difference between Apo-AI mimetic peptides and reconstituted-HDL.

Response: The introduction has now been clarified to address the reviewer’s comment. Please see page 4, line 66-70.

3. In this study, the authors conducted i.p.GTT as an assessment of glucose tolerance, but this method neglected the effects of incretin. Is it not necessary to consider the effect of incretin on the results of this study?

Response: We thank the reviewer for this comment. Intraperitoneal glucose tolerance test in mice followed by measurement of glucose levels in the blood is widely used to give an indication of the presence or absence of diabetes or impairment of glucose tolerance. Whilst we agree that assessment of incretin hormones will value-add to our results, we did not measure incretin levels during our glucose tolerance test in this study but will certainly take this into consideration in our future experimentation. 

4. CETP inhibitor, one of the HDL-targeting drug also improves glucose homeostasis. The authors should consider the difference in the mechanism between the effect of CETP inhibitor and the effect of Apo-AI mimetic peptides.

Response: The discussion now addresses the reviewer’s comment. Please see page 15, line 342-348.

5. HOMA-IR and HOMA-beta should be shown.

Response: We thank the reviewer for this comment. The assessment of IR and B-cell function is commonly used in the clinic. The HOMA model however, has not been validated for use in rodents or any other animals, and such use violates the assumptions of the model. We have therefore chosen not to include these measurements for this study (Ref: Wallace TM, Levy JC, Matthews, DR. Use and Abuse of HOMA Modeling. Diabetes Care 27: 1487-1495, 2004). 

6. Fig. 1A; 4A; 4B; 5A; 6A; 6B, It is very difficult to understand group differences in the line graphs.

Response: To allow for differentiation between the groups, the lines have now been colour coded.

Reviewer #2: Interesting paper examining effect of apoA-I mimetic peptides (D4F and L5F) on glucose metabolism in C57BL6 mice fed a high fat diet.

1. Abstract: Quantitative description of key findings and P-values should be added.

Response: The abstract now includes quantitative description and P-values as requested. 

2. Methods: Method used to quantify visceral and SQ fat should be stated. Time from. Last peptide dose for drawing plasma from mice should be stated.

Response: The method used to quantify visceral and SQ fat has already been included in the methodology section on page 6, line 115-117. The time from last peptide dose have now been added on page 6, line 113-115.

3. Lipoprotein analysis: Ideally, besides plasma triglycerides, total cholesterol should also be analyzed. A more detailed analysis of plasma lipoproteins would also be helpful.

Response: Unfortunately we do not have enough serum for further analysis as suggested. We thank the reviewer for this suggestion and will certainly consider this for future studies. 

4. Discussion: Short paragraph should be added on potential mechanism of action of peptides in regard to observed effects. Oral D4F has been proposed to act via modulating oxidized lipid content in the intestine. This works should be cited and used to possibly explain effect of D4F and lack of effect of L5F. Should discuss how the magnitude of the effect on glucose metabolism by D4F compares to past studies on treatment with rHDL.

Response: A discussion of this has now been included on page 14, line 319 - 326.

---

## [Editor Report · Decision Letter 1]

10 Dec 2019

Apoplipoprotein-AI mimetic peptides D-4F and L-5F decrease hepatic inflammation and increase insulin sensitivity in C57BL/6 mice

PONE-D-19-23129R1

Dear Dr. McGrath,

We are pleased to inform you that your manuscript has been judged scientifically suitable for publication and will be formally accepted for publication once it complies with all outstanding technical requirements.

With kind regards,

Michael Bader

Academic Editor

PLOS ONE
---

## [Editor Report · Acceptance letter]

17 Dec 2019

PONE-D-19-23129R1 

Apoplipoprotein-AI mimetic peptides D-4F and L-5F decrease hepatic inflammation and increase insulin sensitivity in C57BL/6 mice 

Dear Dr. McGrath:

I am pleased to inform you that your manuscript has been deemed suitable for publication in PLOS ONE. Congratulations! Your manuscript is now with our production department. 

With kind regards,

on behalf of

Prof. Michael Bader 

Academic Editor

PLOS ONE